# Influence of Computed Wave Spectra on Statistical Wave Properties

**Tatjana Kokina and Frederic Dias \*** 

School of Mathematics and Statistics, Earth Institute, University College Dublin, Belfield,
D04 V1W8 Dublin 4, Ireland; tatjana.kokina@ucdconnect.ie
\* Correspondence: frederic.dias@ucd.ie

**Abstract:** The main goal of the paper is to compare the effects of the wave spectrum, computed using the Discrete Interaction Approximation (DIA) and the Webb–Resio–Tracy (WRT) methods, on statistical wave properties such as skewness and kurtosis in the context of large ocean waves. The statistical properties are obtained by integrating the three-dimensional free-surface Euler equations with a high-order spectral method combined with a phenomenological filter to account for the energy dissipation due to breaking waves. In addition, we investigate the minimum spatial domain size required to obtain meaningful statistical wave properties. The example we chose to illustrate the influence of the wave spectrum on statistical wave properties is that of a hindcast of the sea state that occurred when the extreme Draupner wave was recorded. The numerical simulations are performed over a physical domain of size 4.13 km × 4.13 km. The results indicate that statistical properties must be computed over an area of at least 4 km$^2$. The results also suggest that selecting a more computationally expensive WRT method does not affect the statistical values to a great extent. The most noticeable effect is due to the energy dissipation filter that is applied. It is concluded that selecting the WRT or the DIA algorithm for computing the wave spectrum needed for the numerical simulations does not lead to major differences in the statistical wave properties. However, more accurate energy dissipation mechanisms due to wave breaking are needed.

**Keywords:** Draupner storm; spectral methods; DIA; WRT; WAVEWATCH III; wave statistics; breaking waves; rogue waves

## 1. Introduction

When studying large ocean waves, the knowledge of statistical wave properties such as skewness and kurtosis allows one to build the probability density function of surface elevations. Nonlinearity leads to a departure of wave field statistics from Gaussianity, which is manifested in skewness and kurtosis [1]. The level of departure could result in disparities in the probability of freak waves. Freak waves can be defined as phenomenal waves, either with their height more than twice the significant wave height (SWH) or with their crest height more than 1.25 times the SWH. The SWH is the mean of the largest third of the waves in a wave record. Rogue waves are rare events, but should not be underestimated, as they can be very dangerous, due to their spontaneous development and the force with which they can impact on marine vessels and structures. Apart from myths or seaman stories, there was no scientific evidence of rogue waves occurrence until 1995. On 1 January 1995, at 15:00, instruments fitted on the Draupner E platform recorded a 25.6 m rogue wave. Since then, the occurrence and prediction of freak waves have been investigated extensively [2–6]. The present study concentrates on the Draupner wave to test the influence of physical and numerical parameters used to obtain the statistical wave properties of large waves.

The first important quantity is the wave spectrum that describes the sea state associated with the occurrence of extreme waves. The concept of sea state is defined as either the temporal domain where the wave field is statistically stationary or the area of the ocean where the wave field is statistically homogeneous. Various methods with different levels of numerical complexity can be used to obtain such spectra. In particular, different methods exist for the computation of the nonlinear quadruplet interactions in the wave spectrum. One of the methods, the Webb–Resio–Tracy (WRT) method [7–9], produces a more accurate physical shape of the power spectrum [10]. Measured power spectra often have a high central peak and enhanced high-frequency channels. In [10], numerical hindcasts of the Draupner storm obtained with the WRT method clearly provide the high-frequency channels. The WRT algorithm is a more detailed algorithm, compared to the discrete interaction approximation (DIA) method [11], as the WRT algorithm computes the full nonlinear interactions. However, being more detailed, the WRT method is also more computationally expensive. Therefore, it is legitimate to check if the fine details of the wave spectrum have consequences on the statistical wave properties when the wave spectrum is used to generate an initial sea state. This is the main goal of the present paper.

After the wave spectrum is used to generate an initial sea state, the evolution of the sea state can be followed by integrating numerically the free-surface Euler equations (water wave equations). Several papers using a high-order spectral (HOS) method have been published recently (e.g., [12–16]). The physical processes responsible for the formation of rogue waves are still under debate. They include nonlinear focusing [17] and dispersive focusing [18,19]. Recent work has provided evidence that real-world rogue waves, at least those under investigation, can be explained without the modulational instability [14]. Despite the debate about the process responsible for the generation of rogue waves, there is agreement on how the probability of these events can be estimated. Using wave statistics it is possible to estimate the probability of rogue wave occurrence. This is an area of active research, and one can find various publications dedicated to this question [15,20–23]. The present study concentrates on two of the most important statistical parameters, namely the skewness and the kurtosis (see Section 2.1 for more details). In general, high values of (excess) kurtosis indicate a greater probability of rogue wave occurrence [14,22,23]. A recent study on the skewness can be found in [24].

In hydrodynamics, truncations of the water wave equations to describe broadband propagation in deep water include the Zakharov Hamiltonian dynamical equations, and the HOS method formulation. These approaches have been shown to be equivalent, at least for weak nonlinearities [5]. The limitations of such approaches are presented in [25]. However, such approaches behave more realistically than approaches based on the nonlinear Schrödinger equation when one deals with real world ocean waves [23]. In this work, we use a HOS method (see Section 4.1 for more details). The initial condition is reconstructed from a hindcast of the Draupner wave, created using the third generation wave model WAVEWATCH III (WW3) with two different energy transfer computation and parameterization methods (Section 4.2). The first question we ask is the following: Does the shape of the spectrum, different due to differences in the production algorithms, significantly affect the statistical values such as skewness and kurtosis? This is an important question, linked to the justification to use WRT to accurately predict statistical wave properties and rogue wave occurrence. Another question of interest is the minimal size of the spatial domain that is required to obtain meaningful values of the statistical properties.

This paper is organized as follows. First, we present the difference between the two wave spectra used as initial conditions for the HOS model. Then, we present the results from the ensemble simulations and the statistical wave properties of interest. Next, we discuss the results and their implications on the probability of occurrence of rogue waves. We also discuss the various choices for the energy transfer computation, for the dissipation due to wave breaking and the spatial domain size. In the Methods Section we present the numerical model used for the simulations and the two different wave spectra used in the study. We also provide a brief description of the statistical wave properties.

## 2. Results

The main objective of this study is to identify differences, if any, in the ocean wave statistics, depending on the initial wave spectra and the different filters for the phenomenological dissipation of wave energy. The initial conditions used for the HOS simulations were obtained from WW3 hindcasts (see the Methods Section for more details).

Depending on the method used for the production of the hindcast of the Draupner storm, the shape of the wave spectrum differs. The two hindcasts used to produce initial conditions for the HOS simulations were produced using the DIA [11] and the WRT model [7–9]. More details can be found in the work of Ponce de León and Osborne [10], who clearly observed two high-frequency channels (bimodality) in their spectrum. What is not clear is whether bimodality is only due to the wave–wave interaction term $S_{nl}$ or it can also be enhanced by the energy input from the wind term $S_{in}$. The analysis of Ponce de León and Osborne [10] does not clearly show the weight of the three terms $S_{nl}$, $S_{in}$ and $S_{diss}$ (energy dissipation due to wave breaking). Figure 1 shows the Draupner wave spectrum produced using WW3 with the DIA method. Figure 2 shows the same spectrum at the same time, but computed using WW3 with the WRT method. Ponce de León and Osborne produced the WW3 spectrum at 14 different 'stations' in the North Sea. In the figures below and throughout the paper, we concentrate on the WW3 spectrum for Station 1, located at $58.17°(58°10'12'')$ north and $2.47°(2°28'12'')$ east for WW3/DIA-40 and WW3/WRT. The third wave spectrum WW3/DIA-30 used in the simulations was produced at location $58.18°(58°10'48'')$ north and $2.47°(2°28'12'')$ east. The actual Draupner E platform, where the instrument that recorded the famous phenomenal wave height is installed, is located at $58.19°(58°11'19.30'')$ north and $2.47°(2°28'0.00'')$ east. In Figure 3, we plot the difference between the two spectra obtained with the two methods mentioned above. The shape of the difference is explained by the fact that peak frequencies and peak directions for the WW3/WRT-40 and WW3/DIA-40 spectra are slightly different. This causes a shift in the direction axis and results in a dip in the plot along $\theta - \theta_p = 60°$. The three-dimensional wave spectrum produced with WRT has a noticeably higher and sharper peak, and the two high-frequency channels are more visible. The higher peak of the WRT spectrum can be explained by the fact that, while the WRT method computes the full nonlinear quadruplet interactions, the DIA method operates with certain approximations. However, we show below that the statistical values obtained with the DIA and WRT methods do not differ significantly enough to claim that bimodality affects rogue waves.

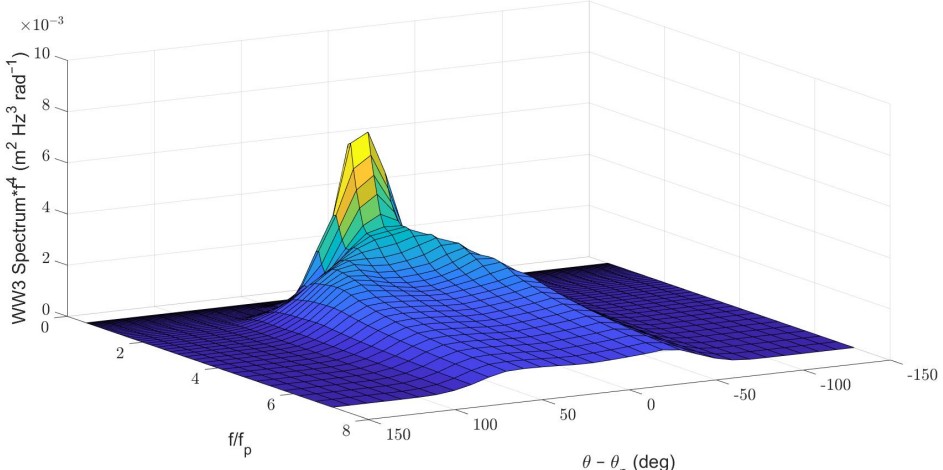

**Figure 1.** Three-dimensional wave spectrum from the WW3/DIA-40 hindcast for the Draupner storm on 1 January 1995 at 15:00, at Station 1, $58.17°(58°10'12'')$N $2.47°(2°28'12'')$E. The spectrum has been transformed to a compensated form via multiplication by $f^4$, where $f$ is the frequency. Frequencies have been made dimensionless by dividing them by the frequency $f_p$ at the peak of the spectrum. The plot is centered around the peak direction $\theta_p$.

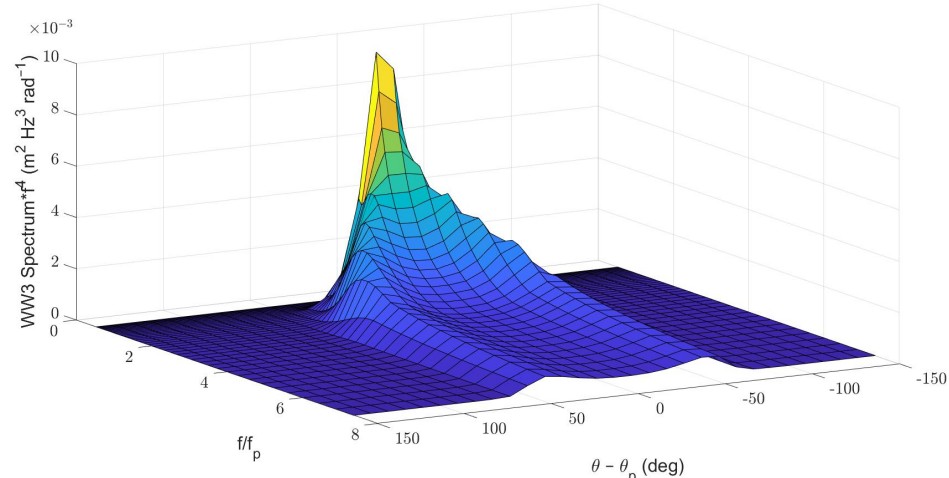

**Figure 2.** Same as Figure 1 for the WW3/WRT-40 hindcast for the Draupner storm on 1 January 1995 at 15:00. The two high-frequency channels are more visible than in the WW3/DIA-40 spectrum.

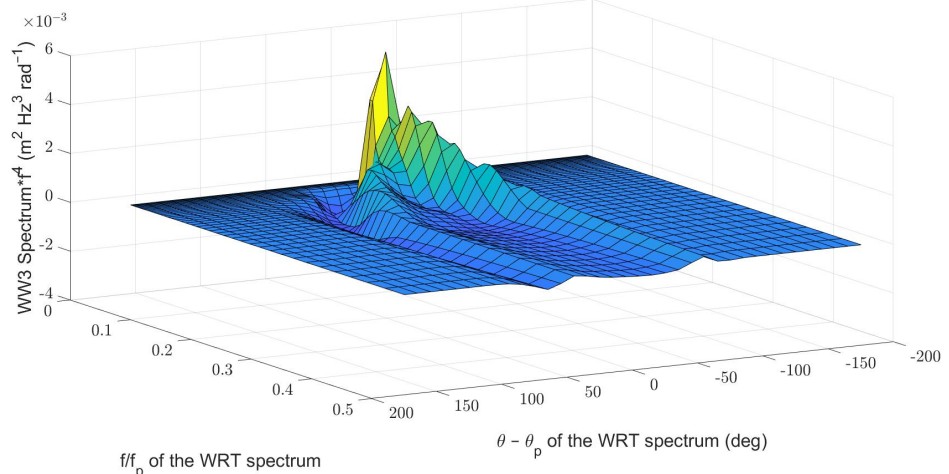

**Figure 3.** Difference between the WW3/DIA-40 and WW3/WRT-40 spectra shown in Figures 1 and 2, respectively, for the Draupner storm.

The plot for the WW3/DIA-30 wave spectrum is not included here as the shape of the DIA-30 spectrum is very similar to that of the DIA-40 spectrum. The two wave spectra plotted above, as well as the DIA-30 spectrum and a different wave spectrum for another event in Ireland on 1 March 2017 (DIA-32), were all used in the HOS simulations. The event of 1 March 2017 belongs to the storm that Met Éireann named Storm Ewan. Note that Ewan was not a particularly strong storm.

### 2.1. Statistical Moments

In this subsection, we present the results of the calculated skewness and kurtosis for both strong and weak filters used for the phenomenological dissipation of wave energy. The results are summarized in Table 1. The statistical values are estimated from an ensemble of 20 HOS simulations each. The calculations take into account the time needed for the development of the nonlinearities, as described in the Methods Section. For the sake of clarity, the 95% confidence intervals are not shown on the figures below. They are similar to the 95% confidence intervals shown in [14,16].

**Table 1.** Kurtosis and skewness average values for an ensemble of 20 HOS simulations for each set up.

| Wave Spectrum | Filter | Kurtosis | Skewness |
|---|---|---|---|
| DIA-30 | Strong | 3.035 | 0.144 |
| DIA-40 | Strong | 3.040 | 0.149 |
| WRT-40 | Strong | 3.038 | 0.156 |
| DIA-30 | Weak | 3.078 | 0.165 |
| DIA-40 | Weak | 3.088 | 0.173 |
| WRT-40 | Weak | 3.096 | 0.182 |
| DIA 1 March 2017 | Strong | 3.002 | 0.092 |
| DIA 1 March 2017 | Weak | 3.051 | 0.130 |

### 2.1.1. Kurtosis for the Draupner Sea State

As mentioned above, the present study pays particular attention to the fourth statistical moment, namely kurtosis. A rogue wave regime is more likely to occur only if the surface spectrum is sufficiently narrow-banded, and it is characterized by a relatively large positive excess kurtosis [14,22,26]. Fedele et al. [14] studied several sets of field data in various European locations with various tools and concluded that the dynamic excess kurtosis is negligible. Thus, third-order quasi-resonant interactions, including modulational instabilities, do not play any significant role in the formation of large waves in comparison to bound nonlinearities especially as the wave spectrum broadens in agreement with oceanic observations available so far. The excess kurtosis is mostly due to bound nonlinearities (see Section 4.4 for more details). Here, it is of particular interest to check whether the wave spectrum (DIA or WRT) that is used influences the values of kurtosis. This has practical implications, as numerical simulations using the WRT method are much more computationally expensive.

In Figures 4 and 5, we show the time evolution of kurtosis calculated from the HOS simulations with strong and weak filters, respectively. The simulations were performed for 30 min of real time (roughly 120 peak wave periods). Time is normalized by the peak wave period $T_p$ of the spectrum. The vertical axis represents the average value of kurtosis over an ensemble of 20 HOS simulations. The strong and weak filters are described in the Methods Section.

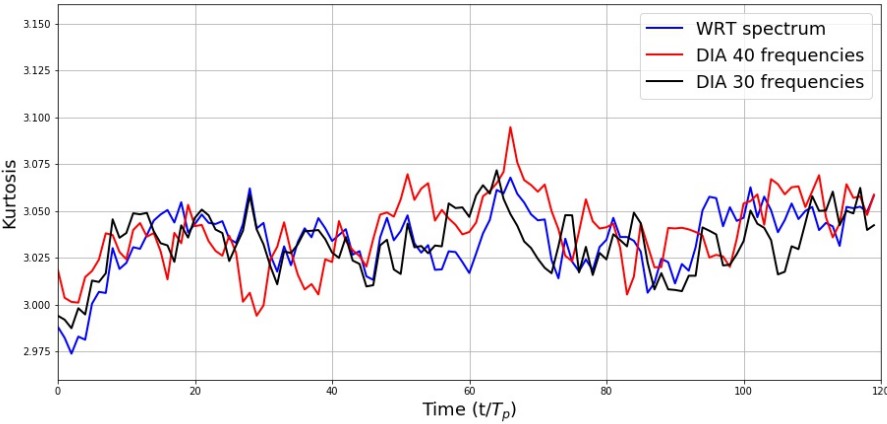

**Figure 4.** Time evolution of kurtosis for the Draupner sea state, for three different wave spectra and the use of the strong energy filter. Time is normalized by the peak wave period $T_p$ of the spectrum. The statistical parameters are estimated from an ensemble of 20 HOS simulations. The initial artificial transients are excluded from the ensemble averages as they are the result of a ramping function applied to the HOS equations to smoothly activate nonlinearities.

Looking at the strong filter simulations (Figure 4), we can see that the overall behavior of kurtosis is similar for the three different input spectra. The numerical values do not deviate from each other.

The initial slight growth of the average kurtosis can be seen for all three spectra in the first $\approx$140 s. Then, the average kurtosis is essentially constant, apart from expected small fluctuations.

The three average values of kurtosis over the whole duration of the simulations that are given in Table 1 are in good agreement. Note that the initial artificial transients are excluded from the averaging process as they are the result of a ramping function applied to the HOS equations to smoothly activate nonlinearities.

In Figure 5, we compare the time evolutions of kurtosis with the application of the weak filter. With the weak filter, we see a much sharper increase in the average kurtosis value in the first $\approx$140 s from values around 3 to 3.1. The computations performed with the WRT spectrum display higher values for kurtosis for the most of the simulation time. The computations performed with the DIA-30 spectrum seem to display slightly lower values of kurtosis. The three average values of kurtosis given in Table 1 show that the weak filter leads to a larger difference between the average values. In future work, it will be interesting to extend the simulations beyond the 30 min of real time to study the behavior of the statistical values during the whole duration of the storm. Note however that if the simulations are run over a longer time, wind forcing must be included. The analysis will be more difficult because stationarity in time and homogeneity in space will no longer be satisfied. Unfortunately, the authors are not aware of any HOS code that can incorporate successfully wind forcing.

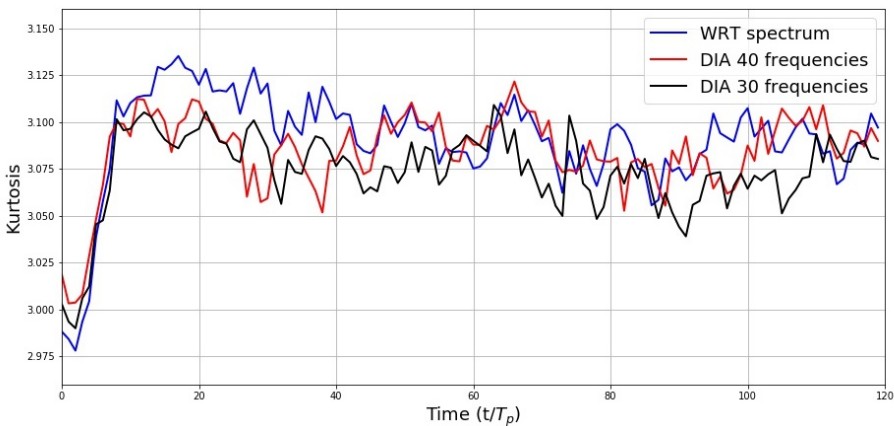

**Figure 5.** Same as Figure 4 with the use of the weak energy filter.

### 2.1.2. Skewness for the Draupner Sea State

Next, we turn our attention to the skewness and perform the same comparisons for the DIA-30, DIA-40, and WRT wave spectra. The time evolution of skewness calculated from the HOS simulations with strong and weak filters, respectively, is presented in Figures 6 and 7, respectively.

In Figure 6, we present the time evolution of the skewness averaged over the ensemble of 20 HOS simulations for the three different input spectra. Time is normalized by the peak period $T_p$ of the spectrum. The first noticeable feature is the sharp increase in the first $\approx$140 s of the simulation. This is again due to the ramping function [27] discussed in the Methods Section. This transient period corresponding to the development of nonlinearities is excluded from the average value calculations presented in Table 1. All three spectra display a similar time evolution. The skewness decreases slowly as time evolves. The trend is the same for all three cases. The values obtained from the simulations with the WRT-40 spectrum are slightly above the other two for the duration of the simulation, with an average value of 0.156.

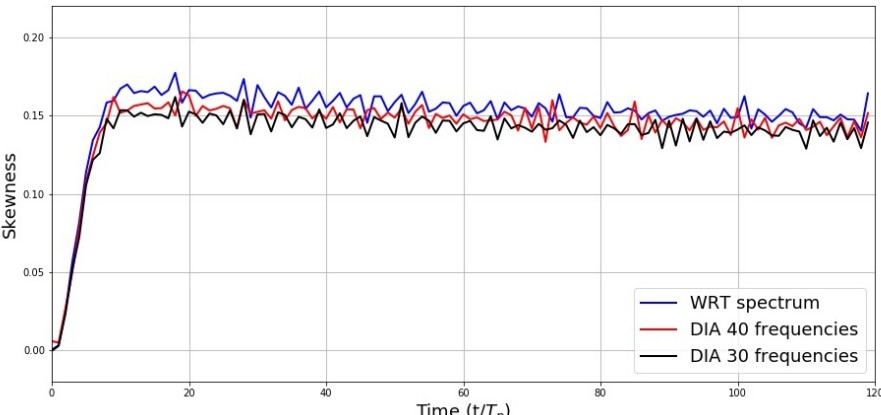

**Figure 6.** Time evolution of skewness for the Draupner sea state, for three different wave spectra and the use of the strong energy filter. Time is normalized by the peak wave period $T_p$ of the spectrum. The statistical parameters are estimated from an ensemble of 20 HOS simulations. The initial artificial transients are excluded from the ensemble averages as they are the result of a ramping function applied to the HOS equations to smoothly activate nonlinearities.

In Figure 7, we repeat the comparisons with the weak filter. The behavior is similar to the one with the strong filter. However, we note higher values of skewness. This is expected, since less energy is dissipated when the weak filter is applied [28]. Again, we see that the values obtained with the WRT-40 spectrum are slightly above the others most of the time, and the difference in the total average values is slightly higher compared to that with the strong filter. The average value for skewness for the simulations using the DIA-30 spectrum is 0.165. This is an interesting result: using the approximation introduced by Tayfun [29], the estimated skewness for the Draupner event is 0.165, which is exactly the same value. Hence, the simulations with the DIA-30 spectrum and the weak filter are in perfect agreement with Tayfun [29].

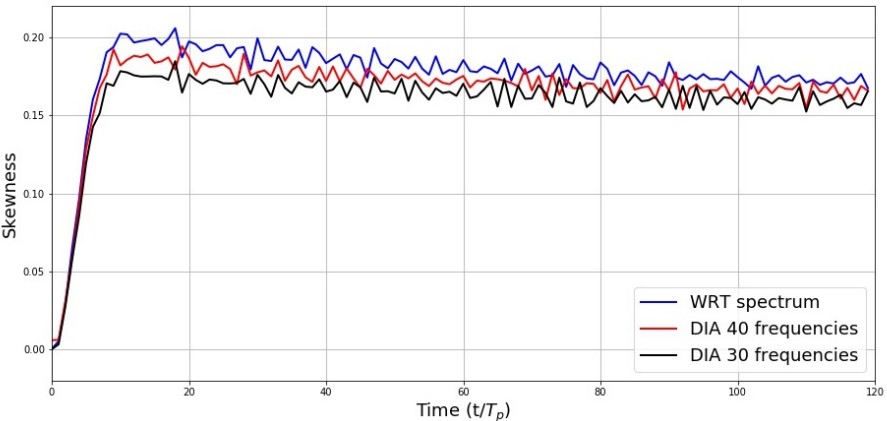

**Figure 7.** Same as Figure 6 with the use of the weak energy filter.

### 2.1.3. Statistical Parameters of the Sea State on 1 March 2017 in Ireland

We also ran the same suite of simulations with a different input spectrum (DIA-32) for a sea state off the west coast of Ireland on 1 March 2017. The coordinates of the location are $54.2856°$ north, $-10.27°$ east. The location is inside the Atlantic Marine Energy Test Site (AMETS), which is being developed by Sustainable Energy Authority of Ireland (SEAI) to facilitate testing of full scale wave energy converters in an open ocean environment. The input spectrum was produced using WW3 again, with a DIA scheme. Details of the wave spectrum can be found in the Methods Section.

The time evolution of kurtosis is shown with the use of both the strong and weak filters in Figure 8. The ensemble averages are presented in Table 1 under DIA 1 March 2017. The general behavior shown in Figure 8 follows the same pattern as in the Draupner sea state: with the use of the weak filter, the kurtosis increases more quickly and the average values for the kurtosis are higher.

The time evolution of skewness is shown in Figure 9. Again, a sharper and higher increase is present in the first $\approx 140$ s of the simulations performed with the weak filter. The values stabilize after 350 s and, as expected, values obtained with the weak filter are higher throughout the whole duration of the simulations.

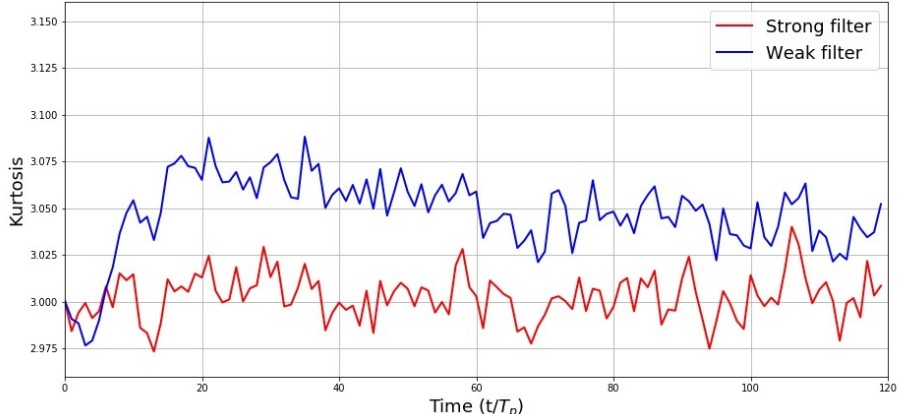

**Figure 8.** Time evolution of kurtosis for the sea state of 1 March 2017, Ireland, with the use of both the strong and weak energy filters. Time is normalized by the peak wave period $T_p$ of the spectrum. The statistical parameters are estimated from an ensemble of 20 HOS simulations.

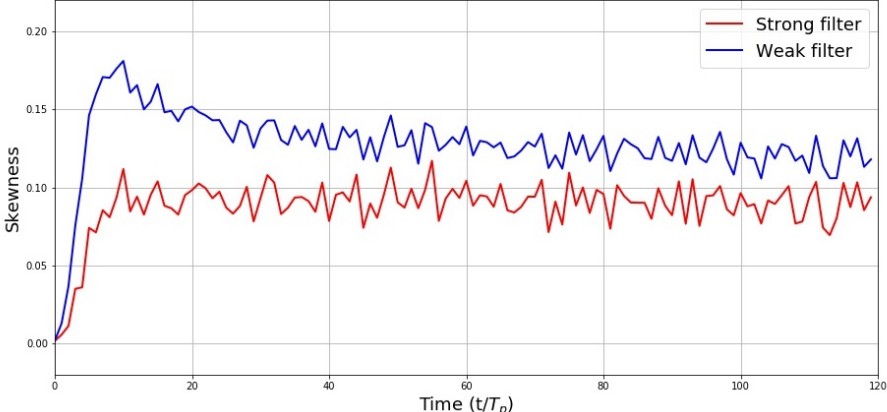

**Figure 9.** Same as Figure 8 for the time evolution of skewness.

## 2.2. Spatial Domain Size

One of the questions we address in the present paper is that of the minimal spatial domain size which is acceptable for a meaningful estimation of kurtosis. Using the DIA-30 strong filter simulations of the Draupner sea state, we calculated the kurtosis starting with a 100 m $\times$ 100 m square and then increasing the size of the domain in steps of 100 m. The values of kurtosis as a function of the size of the domain are shown in Figure 10. The horizontal axis represents the size of the side of the square on which the kurtosis value is calculated. In other words, 100 m, for example, refers to a 0.01 km$^2$ square. The maximum domain size used in the simulations was 4130 m $\times$ 4130 m, or just slightly over 17 km$^2$.

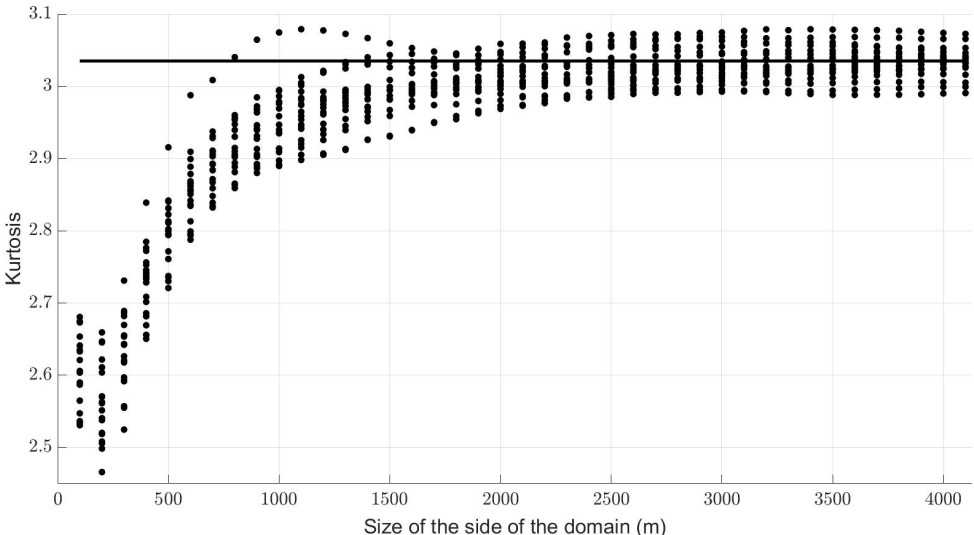

**Figure 10.** Kurtosis as a function of the size of the spatial domain. The horizontal solid line represents the average value of kurtosis from all 20 simulations of the ensemble using the DIA-30 spectrum and the strong filter. Different circles along a given vertical line represent the 20 simulations at each step size.

For a square of 1000 m × 1000 m, none of the simulations reach the value of 3 for the kurtosis except for one outlier. However, as the domain size increases, all simulations seem to converge, and, starting at about 3000 m, the spread is almost uniform. The average simulation values tighten around the average value of all simulations. A zoom-in is given in Figure 11 for square sides ranging from 1000 to 4130 m. In Figure 12, we plot the same results as in Figure 11 with the *x*-axis being a semi-log 1/size of the domain. It appears that the value for mean kurtosis varies linearly as a function of the log of 1/size.

The conclusion is that meaningful values of mean kurtosis are acceptable for squares larger than 2000 m × 2000 m. In other words, the same way that it requires long enough time series to obtain good estimates of mean kurtosis, it requires large enough domain sizes to obtain good estimates of mean kurtosis.

The question of spatial domain size was also considered by Krogstad et al. [30], who presented tools for the spatial extreme value analysis of simulated and measured wave fields. Fedele [31] developed a stochastic approach to model short-crested stormy seas as random fields both in space and time. Defining a space-time extreme as the largest surface displacement over a given sea surface area during a storm, he derived associated statistical properties by means of the theory of Euler characteristics of random excursion sets in combination with the Equivalent Power Storm model. Fedele's space-time model was applied to three real world sea states in [15]. Theoretical ratios between maximum space-time wave height and maximum wave height at a point were plotted as a function of the area width.

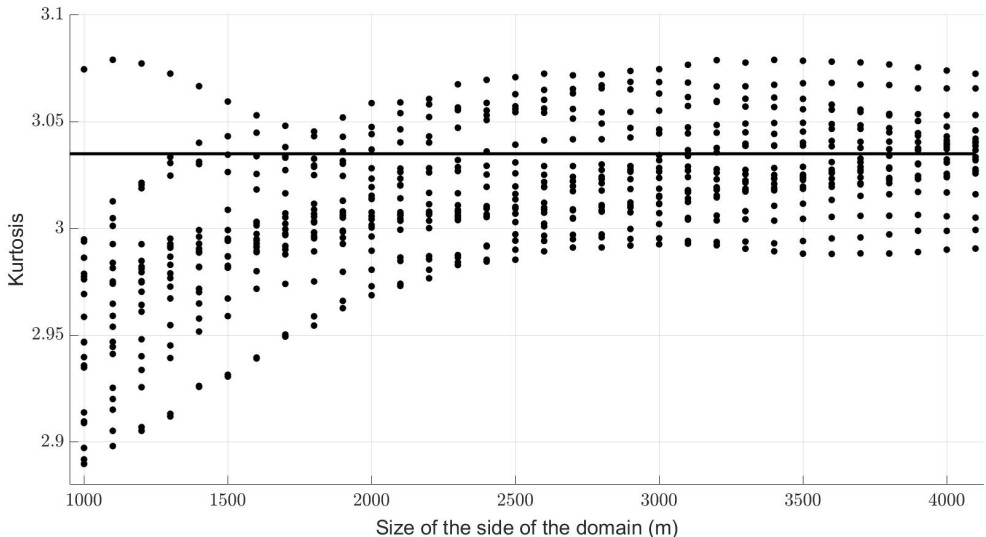

**Figure 11.** Same as Figure 10 with a zoom-in on square sides ranging from 1000 to 4130 m.

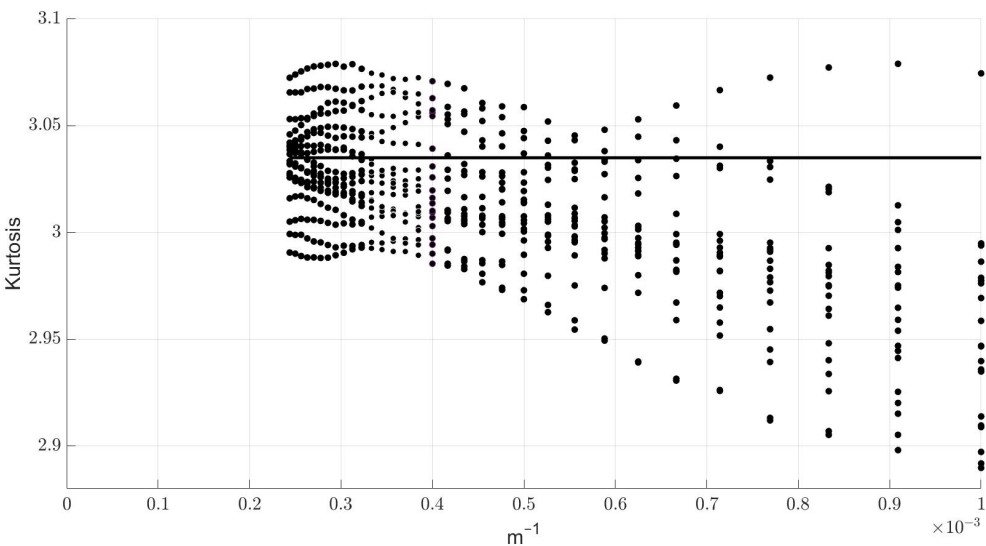

**Figure 12.** Same as Figure 11 with a semi-log plot.

## 3. Discussion

We present statistical wave properties for an energetic event, the Draupner storm, using different input parameters. We investigated the behavior of said properties depending of the wave spectrum, energy dissipation filter, and physical domain size. The question we had in mind was: In a broad sense, what parameters should one use to be able to forecast extreme events accurately, yet with reasonable computational costs?

Two parameterization methods for nonlinear wave energy transfer were considered, and in our results there is no evidence of a critical difference between the two. This can serve as a contribution to reduction in computational costs, since, as mentioned above, WRT is much more expensive, but the statistical parameters obtained from the simulations do not indicate a vital need to use the WRT over DIA method in such simulations. Further reduction in computational costs can be obtained from selecting the appropriate physical domain, and we showed that one should consider the size of the domain carefully, to obtain realistic statistical values. In the particular event we considered (the Draupner storm sea state), the mean wavelength is ≈200 m. If we extrapolate, we could say that

the spatial domain should be at least 10 times the wavelength to obtain meaningful statistical values. It would be interesting to extend this question to the time domain: How long should the simulations be in real time to produce a meaningful result? Fedele et al. [22] determined a so-called optimal sea state duration based on the rationale that variation of key statistics is minimal between two consecutive sea-state sequences. If it is too short, it is meaningless. If it is too long, an implicit assumption of stationarity is made, which is not necessarily satisfied.

The most interesting part of the study is the phenomenological energy dissipation due to breaking and the two filters implemented in the model. As discussed, the HOS method is unable to deal with wave breaking, but we know that breaking is happening, and, to account for the energy dissipation due to breaking, filters are employed. We see how the statistical wave properties increase, if less energy is dissipated, but both filters are an "approximation". They can be good filters, but still do not fully represent breaking waves. Following recent work on wave breaking onset [32] and on wave breaking strength [33], future work will include the inclusion of this breaking onset threshold parameter, to reflect the energy dissipated more accurately. Indeed, it is important to take into account wave breaking as precisely as possible, especially when one tries to forecast rogue waves, since wave breaking reduces wave growth and impedes the occurrence of rogue waves.

We hope our results will motivate further research into the subject, with more accurate wave breaking energy dissipation mechanisms and the inclusion of wind forcing in HOS codes.

## 4. Methods

In this section, we discuss the numerical method, the initial conditions used for the numerical simulations, and the post processing of the simulated data.

### 4.1. Higher Order Spectral Method

In this study, we utilize the higher order spectral (HOS) method that was developed independently by Dommermuth and Yue [34] and West et al. [35] in 1987. The version of West et al. was used in this study. The detailed description of the method can be found in the work by Tanaka [36].

The HOS method is a pseudo spectral method that solves the incompressible Euler equations. Starting with the 3D water-wave problem, the problem is reduced to 2D, by evaluating the quantities of interest on the sea surface. Denoting the two horizontal coordinates as $x$ and $y$ and the vertical coordinate as $z$, the governing equations are

$$\nabla^2 \phi = 0 \qquad\qquad -\infty < z < \eta(x,y,t)\,, \qquad (1)$$

$$\phi_t + gz + \frac{1}{2}(\nabla\phi)^2 = 0 \qquad\qquad z = \eta(x,y,t)\,, \qquad (2)$$

$$\eta_t + \nabla_h\phi.\nabla_h\eta = \phi_z \qquad\qquad z = \eta(x,y,t)\,. \qquad (3)$$

The velocity potential $\phi(x,y,z,t)$ describes the irrotational flow of an inviscid and incompressible fluid, and in the domain of the fluid satisfies the Laplace equation. In addition, at the free surface $\eta(x,y,t)$, the velocity potential satisfies the kinematic and dynamic boundary conditions. Hence, the fluid flow dynamics is described by Equations (1)–(3), where $\nabla_h$ stands for the gradient in the $x - y$ plane.

Introducing the velocity potential on the free surface as

$$\psi(x,y,t) = \phi(x,y,z = \eta,t) \qquad (4)$$

allows to rewrite the boundary conditions at the free surface as follows:

$$\psi_t + g\eta + \frac{1}{2}W^2\left[1 + (\nabla_h\eta)^2\right] = 0,  \tag{5}$$

$$\eta_t + \nabla_h\psi.\nabla_h\eta - W\left[1 + (\nabla_h\eta)^2\right] = 0,  \tag{6}$$

where $W$ is the vertical velocity on the free surface:

$$W = \left.\frac{\partial\phi}{\partial z}\right|_{z=\eta(x,y,t)}.  \tag{7}$$

The main part of the computation is spent solving Equations (5) and (6). Spatial derivatives are computed in spectral space while everything else is computed in physical space. This requires mapping back and forth from physical to spectral space. This is achieved in an efficient way using fast Fourier transforms and their inverse transforms. The simulations were performed using $1024 \times 1024$ Fourier modes. The reason for choosing this particular discretization is based on the convergence results described in Brennan's PhD thesis [28]. Brennan performed simulations with varying grid resolutions: a coarse mesh with resolution $512 \times 512$, a finer $2048 \times 2048$ grid, and the regular $1024 \times 1024$ grid used in the present study. Using the L2 and L1 norms (measured from the free-surface displacement), he showed that there were no significant differences among the coarse, regular, and finer meshes. Therefore, the recommendation was to use the $1024 \times 1024$ grid. In the present study, the spatial domain was set at 4130 m $\times$ 4130 m, which is roughly $20L_m \times 20L_m$ where $L_m$ is the mean wavelength. The simulation time was 1800 s, which is roughly $120T_p$ where $T_p$ is peak wave period. An ensemble of 20 simulations for each set up was chosen. Simulations were subsequently run with an ensemble of 50 simulations for each set up (not shown in the paper because no quantitative differences were noticeable).

During the simulations, the nonlinear terms in Equations (5) and (6) are smoothly activated by the Dommermuth ramping function [27].

As described above, the HOS method is unable to deal with wave breaking, as discontinuous surfaces are not permitted. To account for energy dissipation due to wave breaking, a low pass filter proposed by Xiao et al. [13] is adopted in the simulations, where $k_p$ is the peak wavenumber:

$$F(\mathbf{k}|k_p, f_1, f_2) = \exp\left(-\left|\frac{\mathbf{k}}{f_1 k_p}\right|^{f_2}\right).  \tag{8}$$

Varying the parameters $f_1$ and $f_2$ results in energy dissipation that is in agreement with several numerical and experimental works [37–39]. In this study, two filter set ups were used, $F1 = [8, 30]$ and $F3 = [30, 10]$, which are referred to as strong and weak filters, respectively. The strong filter shows good correlation with experimental work mentioned above (see [28] for more details). However, the weak filter results in a reduced energy loss. We compare the results obtained with applying both strong and weak filters in the Results Section 2.

Initial conditions for the simulations are obtained from the output of WW3, which is described in the following sections.

### 4.2. Energy/Action Balance Equation

We are interested in finding better indicators, or improving the current ones, for rogue wave appearance. To forecast a certain event, we would need to follow the evolution of the sea state from the initial given position. In the evolution of the sea state, we need to be able to account for the effects of energy transfers. To approach this, we want to see how the energy of the wave evolves with time. There are two approaches as how the energy balance of the waves can be formulated—Lagrangian and Eulerian. If we look at the energy evolution equation and differentiate it with respect to time, i.e.,

look at the evolution of the energy equation, this will be equal to a term representing all the effects on energy generation or dissipation, such as wind, wave–wave interactions, and dissipation due to breaking. We can write this equation as

$$\frac{dE(f,\theta;x,y,t)}{dt} = S(f,\theta;x,y,t) \tag{9}$$

where $S(f,\theta;x,y,t)$ is often written as

$$S = S_{in} + S_{nl} + S_{diss} \tag{10}$$

where $S_{in}$ represents the energy input from wind, $S_{nl}$ represents the wave–wave interactions, and $S_{diss}$ is energy dissipation due to wave breaking.

### 4.3. Details of the Wave Spectrum Used in the Simulations

In the simulations, we used four different wave spectra. When we refer to DIA-30, we mean the Draupner storm hindcast produced using the WW3 model, which was developed at NOAA/ NCEP [40,41]. This particular wave spectrum was produced utilizing the DIA method [11] for the computation and parameterization of the nonlinear energy transfer. This model has 36 directional bands and 30 frequencies, with minimum frequency of 0.0350–0.5552 Hz (see [14] for more details).

The DIA-40 wave spectrum is again a Draupner storm hindcast produced using the WW3 operational wave model with the DIA method, and the WRT-40 spectrum represents the same event but with the WRT [7–9] model for the nonlinear energy transfer. More details on the production of the DIA-40 and WRT-40 wave spectra can be found in [10]. Both hindcasts have 36 directional bands (same as DIA-30) and 40 frequencies, with minimum frequency of 0.0350–0.4898 Hz.

The fourth wave spectrum was produced in house, using again the WW3 model. The spectrum was produced using the DIA method. The model has 24 directional bands and 32 frequencies, with minimum frequency of 0.0373–0.71595 Hz.

Let us now review briefly the formulations of the DIA and WRT algorithms. They refer to different treatments of the $S_{nl}$ term, that is the energy transfer due to nonlinear wave–wave interactions. The need for parameterization of the $S_{nl}$ terms arises from the length of time required to compute the term exactly.

One of the existing parameterizations was introduced by Hasselmann [11]—the discrete interaction approximation. In his approach, Hasselmann arrived at the following:

$$\delta S_{nl} = -2\frac{\triangle f\triangle\phi}{\triangle f\triangle\phi}$$
$$\delta S_{nl}^+ = (1+\lambda)\frac{\triangle f\triangle\phi}{\triangle f^+\triangle\phi}$$
$$\delta S_{nl}^- = (1-\lambda)\frac{\triangle f\triangle\phi}{\triangle f^-\triangle\phi}$$
$$\times Cg^{-4}f^{11}\left[F^2\left(\frac{F_+}{(1+\lambda)^4} + \frac{F_-}{(1-\lambda)^4}\right) - 2\frac{FF_+F_-}{(1-\lambda^2)^4}\right]$$

where $C$ is a numerical constant, $\triangle f, f^+, f^-$ denote the discrete resolution of the spectrum and source function at the frequencies $f, f^+$, and $f^-$. The net $S_{nl}$ is calculated by summing over all frequencies, directions, and interaction configurations. For a comparison of the approximate and the exact transfer source function, see [11].

The other parameterization of the $S_{nl}$ term was developed by Webb, Resio, and Tracy [7–9]. Following the mentioned work, one can see that it is beneficial to examine fluxes of action (or energy) past a specific frequency in addition to looking at a source function for the entire spectrum. The energy flux is written as an integral of the density function product with a combination of the Heaviside

functions dependent on wavenumber permutations. Then, the $S_{nl}$ source can be calculated as the one-dimensional divergence of energy flux:

$$S_{nl}(f) = \frac{\partial[\Gamma_E^+(f) + \Gamma_E^-(f)]}{\partial f} \tag{11}$$

(for more details, see [7–9]).

*4.4. Statistical Wave Properties*

Two quantities of particular interest in this study are the skewness $\lambda_3$ and kurtosis (or excess kurtosis) $\lambda_{40}$. These are known as the statistical moments, and are defined as

$$\lambda_3 = \frac{< \eta^3 >}{\sigma^3}, \tag{12}$$

$$\lambda_{40} = \frac{< \eta^4 >}{\sigma^4} - 3. \tag{13}$$

The angle brackets stand for statistical averages and $\sigma$ is the standard deviation of the surface wave elevation.

Plots present in the current work represent full kurtosis, rather than excess. Hence, the plots available in the Results Section 2 show $\lambda_{40} + 3$. As we are dealing with third-order nonlinear random seas, the excess kurtosis $\lambda_{40}$ is made up of two components:

$$\lambda_{40} = \lambda_{40}^d + \lambda_{40}^b, \tag{14}$$

where $\lambda_{40}^d$ represents the dynamic component and $\lambda_{40}^b$ is the bound harmonic contribution. The contribution $\lambda_{40}^b$ comes from the Stokes bound harmonic contribution [42].

The statistical averages are performed over the entire spatial domain (except when we study the influence of the size of the spatial domain for kurtosis) over 20 simulations for each set up. The values presented in Table 1 take into account the time needed for the nonlinearities to develop. This is due to the use of the Dommermuth ramping function [27] mentioned in the above section.

**Author Contributions:** T.K. performed the numerical simulations. T.K. and F.D. wrote the draft article. T.K. and F.D. participated in the analysis and interpretation of results. All authors have read and agreed to the published version of the manuscript.

**Funding:** This research was funded by European Union grant ERC-2018-AdG 833125 HIGHWAVE.

**Acknowledgments:** This work was supported by the European Research Council (ERC) under the research project ERC-2018-AdG 833125 HIGHWAVE. The authors are grateful to Francesco Fedele for suggesting some of the computations performed in this paper, to Sonia Ponce de León for sharing the Draupner wave hindcasts used in this study, to Leandro Fernández for providing the wave spectrum for the sea state of 1 March 2017 and to Joseph Brennan for the development of the HOS code. T.K. acknowledges discussions with Nicole Beisiegel on numerical methods. The authors wish to acknowledge the DJEI/DES/SFI/HEA Irish Centre for High-End Computing (ICHEC) for the provision of computational facilities and support and PRACE for awarding us access to Saga at Sigma2 Metacenter, Norway.

**Conflicts of Interest:** The authors declare no conflict of interest.

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
