# Peer review of "Influence of Computed Wave Spectra on Statistical Wave Properties"

_jmse, doi:10.3390/jmse8121023_

Round 1
Reviewer 1 Report
Review of “Influence of the wave spectrum on statistical wave properties”, by Kokina and Dias This paper examines the influence of the direction/frequency wave spectrum shape on the kurtosis and skewness parameters of the wave field. A set of simulations was based on two different formulations (DIA and WRT) for the nonlinear interactions. Consequence for the extreme wave prediction is commented to some degree. Albeit based on a small set of numerical experiments, the research is genuine and a good starting point for future researches. I have no major issues to point out; some suggestions for improvement are given below for consideration.
- At the beginning of the Introduction, it is stated that “The present study concentrates on the Draupner wave to test the influence of physical and numerical parameters used to obtain the statistical wave properties of large waves.”, while in the abstract all these topics were not mentioned at all. I suggest to point out what the reason is for this study. A clarification would help the reader to follow the flow of the paper.
-Then, the connection between high-order moments and the occurrence of extreme (and rogue) waves may be explained.
- Line 32-33. In the mentioning of nonlinear interactions, authors are concerning the spectral wave models. I’d make it clear (e.g. from observational data, spectra can be computed via Fourier Transform without invoking interactions, whose effect, the other way around, will result in the spectral shape).
-Pg 5, line 111. Is this in contradiction with [13] and Christou and Ewans (2014)? Previously, authors said there is still debate on what mechanism are responsible for the rogue wave occurrence.
-Figure 4 and others following. May authors plot the standard deviation of their estimations? It helps to judge the meaningfulness of the average quantities.
- Pg 6, line 136. If one is concerned with the statistical characterization only, I’d see no issue in making the run longer (> 30 minutes). It would be different if the purpose were to reconstruct the full storm, but that does not seem the case. - Section 2.2. The discussion in this section seem to be connected to the previous studies by Krogstad et al. (1999) and Benetazzo et al. (2016), to which authors are referred to for the lower-order moments of the sea surface elevation distribution.
Reviewer 2 Report
The paper by Tatjana Kokina and Frederic Dias entitled "Influence of the wave spectrum on statistical wave properties" investigates the capabilities of wave models to predict extreme wave events. A particular emphasis was given to the effect of initial spectrum shape, spatial size of the domain, and empirical closures of the models. The manuscript is very well written and addresses an important practical and fundamental problem of oceanography. I would like to strongly encourage the publication of the paper.
Several minor questions appeared while reading the paper:
- Is it possible to study the given problem by algorithms other than HOS method? For instance, by methods purely formulated in physical space.
- The problem of domain size is discussed in the paper. Is it also important to consider both temporal and spatial discretization? It is indicated that 1024x1024 Fourier modes are used. What happens if the resolution is reduced or refined?
- From the reviewer's point of view, it is of interest to provide the domain recommended size in dimensionless variables facilitating easy scaling of the problem to other sea states. Is it possible to provide some comments on this point?
